# Glioblastoma Cells Counteract PARP Inhibition through Pro-Survival Induction of Lipid Droplets Synthesis and Utilization

**DOI:** 10.3390/cancers14030726

**Published:** 2022-01-30

**Authors:** Jara Majuelos-Melguizo, José Manuel Rodríguez-Vargas, Nuria Martínez-López, Daniel Delgado-Bellido, Ángel García-Díaz, Víctor J. Yuste, Marina García-Macía, Laura M. López, Rajat Singh, F. J. Oliver

**Affiliations:** 1Department of Immunology and Cell Biology, Institute of Parasitology and Biomedicine “López-Neyra”, IPBLN, CSIC, CIBERONC, 18016 Granada, Spain; jara_mm@hotmail.com (J.M.-M.); ddelgado@ipb.csic.es (D.D.-B.); agdiaz@ipb.csic.es (Á.G.-D.); laural@ipb.csic.es (L.M.L.); 2Department of Biochemistry, Autonomous University of Madrid (UAM), ‘Alberto Sols’ Biomedical Research Institute (CSIC-UAM), 28029 Madrid, Spain; 3Institute for Cell and Molecular Biosciences, Newcastle University, Newcastle upon Tyne NE2 4HH, UK; nuria.martinez-lopez@einsteinmed.org; 4Cell Death, Senescence and Survival Group, Department of Biochemistry and Molecular Biology and Institute of Neurosciences, Faculty of Medicine, Autonomous University of Barcelona, CIBERNED, 08193 Barcelona, Spain; victor.yuste@uab.cat; 5Institute of Cellular Medicine, Newcastle University, Newcastle upon Tyne NE2 4HH, UK; marinagarciamacia@usal.es; 6Department of Medicine, Albert Einstein College of Medicine, Bronx, NY 10461, USA; rajat.singh@einsteinmed.org; 7Department of Molecular Pharmacology, Albert Einstein College of Medicine, Bronx, NY 10461, USA

**Keywords:** glioblastoma stem cells, PARP inhibitors, lipophagy, lipid droplets, Acyl-coA-carboxylase, metabolic adaptation

## Abstract

**Simple Summary:**

Glioblastoma multiforme (GBM) is the most common and deadly primary brain tumor in adults and one of the most aggressive cancers. The use of Poly ADP-Ribose Polymerase (PARP) inhibitors is being expanded as therapeutic alternative in multiple types of cancer beyond BRCA1/2 mutant breast and ovarian cancer. Here we have analyzed glioma cells’ traits that limit the efficacy of PARPi as anti-glioma agents and we found that PARPi triggered the synthesis of lipid droplets (LDs) that fueled glioma cells by inducing pro-survival lipid consumption. Notably, blocking Fatty Acids utilization by inhibition of β-oxidation with etomoxir, increased PARPi-induced glioma cell death while treatment with oleic acid (OA) prevented the anti-glioma effect of PARPi. We uncover a novel mechanism by which glioblastoma escapes to anti-tumor agents through metabolic reprogramming, inducing the synthesis and utilization of LDs as a pro-survival strategy in response to PARP inhibition.

**Abstract:**

Glioblastoma multiforme (GBM) is the most common and aggressive primary brain tumor in adults. Poly (ADP-ribose) polymerase inhibitors (PARPi) represent a new class of anti-neoplastic drugs. In the current study, we have characterized the mechanism by which glioblastoma cells evade the effect of PARPi as anti-tumor agents. We have found that suppression of PARP activity exerts an anti-stemness effect and has a dual impact on autophagy, inducing its activation in the first 24 h (together with down-regulation of the pro-survival mTOR pathway) and preventing autophagosomes fusion to lysosomes at later time-points, in primary glioma cells. In parallel, PARPi triggered the synthesis of lipid droplets (LDs) through ACC-dependent activation of de novo fatty acids (FA) synthesis. Notably, inhibiting β-oxidation and blocking FA utilization, increased PARPi-induced glioma cell death while treatment with oleic acid (OA) prevented the anti-glioma effect of PARPi. Moreover, LDs fuel glioma cells by inducing pro-survival lipid consumption as confirmed by quantitation of oxygen consumption rates using Seahorse respirometry in presence or absence of OA. In summary, we uncover a novel mechanism by which glioblastoma escapes to anti-tumor agents through metabolic reprogramming, inducing the synthesis and utilization of LDs as a pro-survival strategy in response to PARP inhibition.

## 1. Introduction

Glioblastoma multiforme (GBM) is the most common primary malignant brain tumor in adults. In spite of some evident progress in strategies against GBM, prognosis remains extremely poor with the average survival reported to be less than 15 months after diagnosis. The nuclear protein PARP-1 functions as a DNA damage sensor and is also implicated in a broad variety of cellular functions, including metabolic adaptations [1,2]. PARP inhibitors exhibit anti-tumor activity in part due to their ability to induce synthetic lethality in cells deficient for homologous recombination repair [3,4,5,6]. Up to date, the studies on GBM and PARPi have focused on the use of these small molecules as radio or chemo-potentiators [7,8]. PARPi enhance the efficacy of Temozolomide (TMZ) in multiple preclinical models and are a promising strategy for GBM [9,10,11]. In a previous study we have reported that PTEN-deficient glioblastoma cells were particularly sensitive to PARPi because they aggravated homologous recombination repair deficiency and displayed genomic instability leading to mitotic catastrophe [12]. In the current study, we have analyzed glioma cells’ traits that limit the efficacy of PARPi as anti-tumor agents. Emerging evidences demonstrate that lipid metabolism undergoes reprogramming in cancer cells in general and in glioma in particular [13,14]. Here we show that PARPi induces the formation of LDs that serve as fuel for tumor cells, which occurs in parallel with the induction of autophagy. Mechanistically, PARPi-induced LDs formation occurs through synthesis de novo, dependent on the activation of Acyl-CoA-Carboxylase after AMPK suppression as a consequence of PARPi-induced ATP recovery. This mechanism occurs concomitantly with the increase in the transcriptional activation of fatty acid synthase (FASN). Moreover, preventing lipid synthesis and utilization by glioma cells increased PARPi-induced glioma cell death suggesting that avoidance of metabolic reprogramming might result in potentiation of PARPi antitumor efficacy.

## 2. Materials and Methods

### 2.1. Cell Culture and Reagents

U87MG (PTEN mutant) cell line was purchased from the European Collection of Authenticated Cell Cultures (Cat. 89081402) through the University of Granada Cell Lines Service. LN929 (PTEN wild type) cell line was a donation from Dr. Joan Seoane (Vall d’Hebron Institute of Oncology, Barcelona, Spain). SW1783 (PTEN mutant) grade III astrocytoma cell line and MSO4 (mutations not characterized) primary glioblastoma cell line, were cultured in high-glucose Dulbecco’s Modified Eagle’s Medium (DMEM) supplemented with 10% heat-inactivated fetal bovine serum (FBS, Gibco), 100 mg/L Penicillin and 500U Streptomycin (Sigma-Aldrich, St Louis, MO, USA, P4333). Patient-derived Glioma Stem Cells TG1A and TG16 (PTEN wild type) were kindly provided by Hervé Chneiweiss and Marie-Pierre Junier’s lab in Quai Saint-Bernard, Paris, and they were obtained as described previously [15] and cultured in DMEM/F12 supplemented with N2, G5 and B27 (Invitrogen, Waltham, MA, USA). All cells were maintained at 37 °C and 5% CO_2_. PJ34 ([*N*-(6-Oxo-5,6-dihydrophenanthridin-2-yl)-(*N*,*N*-dimethylamine) acetamide hydrochloride]) (Alexis Biochemicals, Barcelona, Spain, ALX-270-289), AZD2281/Olaparib (Selleckchem, Houston, TX, USA, S1060) and BMN 673 (Selleckchem, S7048) were used as PARP inhibitors. Olaparib was dissolved in DMSO, and PJ34 and BMN 673 were dissolved in H2O. Cells were treated at the indicated doses and times. mTORC1 inhibitor Rapamycin (Calbiochem, St. Louis, MO, USA, #553210) was dissolved in ethanol and used at a final concentration of 100 nM. Chloroquine (Sigma-Aldrich, C6628), NH_4_Cl (American Bioanalytical, New York, NY, USA, AB00161) and Leupeptin (Fisher, Pittsburgh, PA, USA, BP2662100) were used to inhibit lysosomal degradation activity. They all were dissolved in H_2_O. Chloroquine was used at a final concentration of 10 μM. NH_4_Cl was used at a final concentration of 20 mM and Leupeptin at 100 µM. AMP analog AICAR (Sigma-Aldrich, A9978) was dissolved in H_2_O and used at a final concentration of 2 mM. The ROS inhibitor N-acetyl cysteine (NAC) (Sigma-Aldrich A7250) was dissolved in H_2_O and used at a final concentration of 3 mM. β-oxidation inhibitor Etomoxir (Sigma-Aldrich, E1905) was dissolved in H_2_O and used at a final concentration of 100 μM. Oleic acid (Sigma-Aldrich, O3008) was used at a final concentration of 0.25 mM and was added 12 h before PJ34 treatment. In all cases, cells were plated 24 h before the treatments.

### 2.2. Western Blot Analysis

Cells were plated in 6-well plates at a density of 1.5 × 10^5^ cells per well. After the treatments, cells were washed twice with 1× Phosphate Buffered Saline (PBS) and resuspended in lysis buffer RIPA (150 mM NaCl, 1% NP40, 0.5% sodium deoxycholate, 0.1% SDS, 50 mM Tris pH 8), supplemented with protease and phosphatase inhibitors. Lysates were sonicated, centrifuged at maximum velocity for 30 min at 4 °C and the supernatant was collected. Protein quantification was determined using the Lowry assay (Bio-Rad laboratories, Hercules, CA, USA, #500-0112). Proteins were resolved on SDS-polyacrylamide gels and transferred onto PVDF Membrane (Bio-Rad laboratories). Membranes were blocked with 5% nonfat dry milk powder in TBST (1× TBS containing 0.1% Tween-20) for 1 h and incubated overnight with 1% milk TBST with the following antibodies: Anti-PARP-1 (C2-10 mouse, ALEXIS, LA, ALX-804-210), anti-PAR (10H) (Millipore, Burlington, MA, USA, MABC547), anti-P-PTEN (S380) (Cell Signaling Technology, Danvers, MA, USA, #9551), anti-PTEN (A2B1) (Santa Cruz Biotechnology, Dallas, TX, USA, sc-7974), anti-P-AKT (Ser473) (Cell Signaling Technology, #9271), anti-AKT (Cell Signaling Technology, #4691), anti-P-S6 (Ser235/6) (Cell Signaling Technology, #2211), anti-P-GSK3β (Ser9) (Cell Signaling Technology, #9336), anti-β-CATENIN (Sigma-Aldrich, C2206), anti-LC3B (Cell signaling Technology, #2775), anti-SOX2 (Millipore, AB5603), anti-OCT4 (Abcam, Cambridge, UK, ab18976), anti-NANOG (Millipore, PA1-097) anti-P-P70^S6k^ (Thr389) (Cell Signaling Technology, #9205), anti-P70^S6k^ (Cell Signaling Technology, #9202), anti-ILK (Millipore, MABT66), anti-P-ACC (Ser79) (Millipore, 07-303), anti-P-AMPK (Thr172) (Cell Signaling Technology, #2535S), anti-AMPK (D5A2) (Cell Signaling Technology, #5831), anti-ATG1 (Cell Signaling Technology, #4776). Anti-α-Tubulin (Santa Cruz Biotechnology, sc-8035), anti-GAPDH (Sigma-Aldrich, St G9545) and anti-β-Actin (Sigma-Aldrich, A5316) were used as loading control. Bands were visualized with AmershamTM ECLTM Western Blotting Detection Reagent (GE Healthcare, Chicago, IL, USA, RPN2106) and ImmobilonTM Western (Millipore, WBKLS0500). Pictures were taken with the imaging system ChemiDoc XRS System (Bio-Rad) or medical X-ray films (AGFA, Mortsel, Belgium). To quantify western blot signals we used ImageJ and all the films were exposed for different times to make sure that the bands are in the linear range of detection. In all the cases we compare the relative abundance of a specific protein (Bands) based in two methods for normalizing WBs: (i) housekeeping protein normalization α-tubulin or β-actin; (ii) total protein normalization, total Akt, AMPK, P70^S6k^. In all the cases, 3 independent experiments were migrated to highlight the results. Most representative’s gels of each experiment have been shown including molecular weight according with antibodies data sheet.

### 2.3. Transfection of Small Interfering (si) RNA (siRNA)

Cells were plated in 6-well plates at a density of 1 × 10^5^ cells per well; 24 h later, cells were transfected with the indicated siRNAs at 50 nM using Lipofectamine^®^ 2000 (Invitrogen, 11668) or Jet PRIME (Polyplus Transfection^®^, Alsace, France, 114-75), according to the manufacturer’s guide. Double-stranded RNA duplexes corresponding to a non-targeted control (scramble) (5′-CCUACAUCCCGAUCGAUGAUGUU-3′) and ATG1 (5′-CAGCAUCACUGCCGAGAGGUU-3′, 5′-CCACGCAGGUGCAGAACUAUU-3′, 5′-GCACAGAGACCGUGGGCAAUU-3′, 5′-UCACUGACCUGCUCCUUAAUU-3′) were ordered to Sigma-Aldrich. Double-stranded RNA duplexes corresponding to human PARP1 were from Ambion Applied Biosystems (Burlington, MA, USA), and double-stranded RNA duplexes corresponding to human ACC were from Thermo Scientific (Burlington, MA, USA) (siRNA against ACACA #s883; siRNA against ACACB #s887). 48 h after transfection, cells were treated with the correspondent treatment as indicated above.

### 2.4. Transfection of mCherry-GFP-LC3

Cells were plated in 6-well plates at a density of 1 × 10^5^ cells per well on glass cover-slips. 24 h later, transfection was performed with 0.5 μg mCherry-GFP-LC3 plasmid (kindly provided by Dr S. Hilfiker (Department of Immunology and Cell Biology. Institute of Parasitology and Biomedicine “López-Neyra”, IPBLN, CSIC, Granada, Spain) and ideated by Dr. T. Johansen (The Arthic University of Norway, Tromsø, Norway)) using Jet PRIME (Polyplus Transfection^®^, 114-75) according to the manufacturer’s protocol. Cells were treated 48 h after the transfection. After the treatments, cells were fixed with paraformaldehyde (PFA) solution (4% PFA (*w*/*v*), 2% (*w*/*v*) sucrose in 1× PBS) for 10 min at room temperature and permeabilized with 1× PBS containing 0.5% Triton X-100 for 5 min at room temperature. Nuclei were stained with DAPI and immunostaining was visualized with Confocal Leica LCS SP5 (Wetzlar, Hesse, Germany) using LAS AF software. Quantification of the yellow and red puncta was performed using at least five fields of view (at least 150 cells) per condition in three independent experiments.

### 2.5. Autophagy Flux Assay

Cells were plated in 6-well plates at a density of 1.25 × 10^5^ cells per well. Cells were treated with in presence or absence of lysosomal inhibitors (Leupeptin (200 mM) and ammonium chloride (20 mM)) for 4 h. Net autophagy/LC3-II flux was determined by subtracting the densitometric value of Inh-untreated LC3-II from corresponding Inh-treated LC3-II value. Steady state LC3 was calculated as LC3-II/LC3-I ratio. Rate of fusion is the result of dividing the densitometric value of Inh-treated LC3-II by corresponding Inh-untreated LC3-II value. All results were normalized to the correspondent control in 3 independent experiments.

### 2.6. RT-qPCR

Primers were designed using NCBI primers design software as follows: human FASN→Forward→5′-ACCAGAGCAGCCATGGAGGAG-3′; →Reverse→5’-CGTAGAGCCCCGCCTTCCAG-3′.

Cells were plated in 6 wells at a density of 10^5^ cells. RNA extraction was carried out using RNeasy MiniKit (Qiagen Inc., Venlo, The Netherlands) and genomic DNA was eliminated via gDNA Eliminator spin columns. RNA quantification was performed using NanoDROP© ND-1000 Spectrophotometer (Thermo Fisher). Complementary DNA or cDNA obtaining was performed with Invitrogen M-MLV Reverse Transcriptase (Invitrogen #10338842) and qPCR was performed using Power SYBR Green PCR Master Mix (Applied Biosystems, Waltham, MA, USA) on a StepOnePlus Real-Time PCR System (Applied Biosystems). Expression of Fatty acid synthase (FASN) was normalized to TATA-binding protein (TBP) and all reactions were performed in triplicate.

### 2.7. GICs Immunofluorescence

Cells grown as neurospheres were dropped on poly d-lysine-coated slides (Microscope slide Polysine adhesion Thermo Scientific J2800AMNZ) and fixed in PBS-PFA 4%. Next, cells were permeabilized with 1× PBS containing 0.5% Triton X-100. SOX2 and NESTIN (Millipore) were used at a dilution of 1:100, and nuclear counterstaining with DAPI was performed after removal of excess secondary antibody. Immunostaining was visualized with Confocal Leica LCS SP5. At least 250 cells in 10 independent fields of view per condition and replicate were analyzed in 3 independent experiments.

### 2.8. LAMP1-Bodipy Immunofluorescence

2.5 × 10^4^ cells were seeded on poly d-lysine-coated coverslips in 12-well plates (Sigma-Aldrich, P0899). After the treatments, cells were fixed with 4% PFA in 1× PBS for 30 min at room temperature and washed three times in 1× PBS. Cells were permeabilized with 0.5% Triton X-100, blocked for 1 h and incubated with primary antibody against LAMP1 (Abcam, #241170) and secondary antibody (Alexa Fluor 647 conjugated) (Invitrogen). For lipid droplet detection, cells were incubated with BODIPY 493/503 (Molecular Probes, Eugene, OR, USA, D-3922) for 20 min at room temperature. Coverslips were mounted using mounting medium contained DAPI (4′,6-diamidino-2-phenylindole) (Invitrogen). Immunostaining was visualized with Leica DMi8 wide field fluorescence (Inverted) microscope. Images were deconvolved using the Huygens software (https://svi.nl/Huygens-Software, accessed on 12 November 2021). All images were subjected to identical post-acquisition processing. Percentage colocalization was calculated using the JACoP plugin in single Z-stack sections of deconvolved images using at least 10 fields of view per condition in 2 independent experiments.

### 2.9. Sudan-Red Staining

Cells were grown in 12 well-plates at a density of 2.5 × 10^4^ cells per well on glass coverslips. After the treatments, cells were fixed with PFA solution (4% PFA (*w*/*v*), 2% (*w*/*v*) sucrose in 1× PBS) for 10 min at room temperature and permeabilized with 1× PBS containing 0.5% Triton X-100 for 5 min at room temperature. Next, lipid droplets were labeled with 0.2% Sudan III for 20 min at room temperature. Nuclear counterstaining with DAPI was performed after removal of excess of the lipidic dye. Immunostaining was visualized with Confocal Leica LCS SP5 Fluorescence Microscope using LAS AF software. Quantification of lipid droplets was performed with ImageJ in at least 150 cells, using at least 5 fields of view per condition in 3 independent experiments.

### 2.10. ATP Determination

Cells were grown in 6 well-plates at a density of 5 × 10^5^ cells. Intracellular ATP was measured using a luciferin/luciferase-based assay (Adenosine 5′-triphosphate (ATP) Bioluminescent Assay Kit, Sigma-Aldrich, FLAA-1KT) following manufacturer’s instructions. Protein concentration was determined using Lowry protein assay reagent (Bio-Rad). The content of ATP was normalized to protein content and expressed as percentage of control.

### 2.11. Short-Term Proliferation Assay: MTT

For MTT (3-(4,5-Dimethylthiazol-2-yl)-2,5-diphenyl Tetrazolium Bromide) assay, cells were grown in 96 well-plates at a density of 8 × 10^3^ cells. MTT assay was performed using Cell Proliferation Kit I (MTT, 1-65-007, Roche, Mannheim, Germany) following manufacturer’s instructions. Plate absorbance was read at a wavelength of 570 nm in VERSAmax plate reader (Molecular Devices, St José, CA, USA).

### 2.12. SubG1 Analysis: Propidium Iodide

Cells were grown in DMEM without FBS in 6 well-plates at a density of 1.5 × 10^5^ cells. After the indicated treatments, cells were trypsinized, washed with 1× PBS, permeabilized with 70% ice cold ethanol for at least 10 min, washed again with 1× PBS and incubated with Propidium iodide 40 μg/mL and RNAase A 100 μg/mL (Ribonuclease A from bovine pancreas, Sigma-Aldrich, R6513-10MG) at 37 °C in dark for 20 min. Cells were analyzed on a FACScalibur using CellQuest software (Becton Dickinson, Mountain View, CA, USA) and cell cycle was determined using FlowJo software.

### 2.13. Seahorse Respirometry

Cells bioenergetics was determined using a Seahorse respirometer (Seahorse Bioscience, North Billerica, MA, USA). Briefly, cells were grown in a XF96 plate at a density of 7.5 × 10^3^ cells. Treatments with PJ34, lysosomal inhibitors and etomoxir were performed as indicated. One hour before the end of the experiment, DMEM was replaced by freshly prepared Seahorse Assay Medium XF base medium (Sigma-Aldrich D5030), 5 mM Glucose, 2 mM L-glutamine (GIBCO 35050), 1 mM Sodium pyruvate (GIBCO 11360), Phenol Red (Sigma-Aldrich P5530) and incubated in a CO_2_-free incubator for 1 h. 1.25 µM Oligomycin (Sigma-Aldrich 75351), 2 µM FCCP (Sigma-Aldrich C2920), 1.8 µM Antimycin (Sigma-Aldrich A8674) were then added as indicated, and basal oxygen consumption rates (OCR) values were normalized to protein quantification using the Lowry assay (Bio-Rad laboratories, #500-0112).

### 2.14. Transmission Electron Microscopy

Cells were grown in 10 cm plates at a density of 1 × 10^6^ cells. After the treatments, cells were trypsinized, washed with 1× PBS and fixed in fixation solution (2% Glutaraldehyde, 1% Formaldehyde in 0.05 M Cacodilate buffer pH 7.4) for 5 h. Next, cells were incubated three times in washing solution (Cacodilate 0.1 M pH 7.4) for 15 min. Finally, samples were stained with uranil acetate (Uranyl Acetate 2% in miliQ H_2_O). The ultrathin sections were performed with a diamond knife in an ultra-microtome (Reichert Ultracut S, Leica Microsystems, Wetzlar, Hesse, Germany). The samples were analyzed in a TEM Zeiss 902 (Jena, Turingia, Germany) with 80 Kv of voltage acceleration (CIC-UGR, Granada, Spain).

### 2.15. Statistical Analysis

Independent experiments were pooled when the coefficient of variance could be assumed identical. Statistical significance was evaluated using Student’s *t* test (*n* = number of independent experiments). *P* values below 0.05 were considered significant. (* *p* < 0.05, ** *p* < 0.01, *** *p* < 0.001, # *p* < 0.05, ## *p* < 0.01).

## 3. Results

### 3.1. PARP Inhibition Down-Regulates Pro-Survival Pathways in Glioblastoma

The presence and involvement of Glioma Stem-like Cells (GSCs), also named Glioma Initiating Cells (GICs), in the initiation and propagation of GBM is broadly accepted [16], and the comprehension of their basic biology is critical to understand tumor relapse and failure of treatments. Currently, any anti-tumor approaches against GBM must necessarily target, one way or another, GSCs population. Consequently, to fully recognize the effect of PARPi against GBM we aimed to evaluate their action on GSCs. For this purpose, we used to patient-derive primary glioma stem-like cells TG1A and TG16 [15]. Due to the intrinsic resistance of GSCs to different treatments, we exposed cells to PARPi for one week and re-added PARPi during the first three days of the experiment to counteract the effect of surface ABC transporters [17].

We first analyzed the activity of the AKT/mTOR axis after PARP inhibition. Treatment with either PJ34 or Olaparib resulted in AKT/mTOR pathway down-regulation (Figure 1A). We observed decreased phosphorylation of PTEN (Ser380), which reflects PTEN activation and consequently constitutes a brake for AKT/mTOR signaling. In addition, we detected decreased phospho-AKT (Ser473) levels accompanied by diminished phospho-GSK3β (Ser9) and β-catenin levels, both downstream effectors of AKT. mTOR inactivation was further confirmed by decreased phosphorylation of the mTOR signaling target ribosomal protein P70S6k (Thr389) and increased LC3 II/I levels (Figure 1A).

As the mTOR cascade in GSCs is involved in the maintenance of the stem phenotype [18,19], the results observed in Figure 1A prompted us to analyze the status of stem markers such as SOX2, Oct4 and Nanog following PARPi. A consistent decrease was detected at the protein level, measured by western blot (Sox2) (Figure 1B) and immunofluorescence (Sox 2, Nestin) (Figure 1C).

To get more mechanistic information on the role of PARP, we examined the status of mTOR pathway following PARP inhibition with PJ34 or Olaparib in two GBM cell lines (LN229 and U87MG). The efficiency of both PARP inhibitors in abrogating poly ADP-ribose (PAR) synthesis was confirmed by western blot of PAR in LN229 and U87MG GBM cell lines (Appendix A).

A significant decrease in the phosphorylation of AKT or ILK as well as the mTOR substrate P70S6K were observed after treatment with PJ34 (Figure 2A and Appendix A).

Treatment with Olaparib elicited similar results, promoting the down-regulation of the mTOR activator phospho-AKT and phospho-S6 (Appendix A). Next, we tested transient PARP-1 knock-down and we observed that the effect was not as pronounced as with PARP inhibition respect to phospho-AKT, although increased LC3-II/LC3-I levels as well as decreased P70^S6k^ phosphorylation (Figure 2B and Appendix A), thus supporting that PARP-1 is responsible for this effect, and discarding any off-target effect of PARP inhibitors. Then, we performed Transmission Electron Microscopy (TEM) to evaluate ultra-structural morphological changes. As observed in Figure 2C, both Olaparib and PJ34 induced the accumulation of autophagic vesicles. Interestingly, basal autophagy was also detected (especially in U87MG cells) (Figure 2C and Appendix A).

To confirm that PARPi induced autophagy in LN229 cells we assessed autophagosomes formation and their clearance by lysosomes, we transfected cells with the mCherry-GFP-LC3 reporter (fusion of pH-insensitive mCherry, pH-sensitive green fluorescent protein and LC3) that emphasizes autophagosomes as yellow puncta and autolysosomes (post-lysosomal fusion) as red puncta [20]. This pattern estimates the number of each intermediate; if autophagic flux increases, both yellow and red puncta are increased. If, instead, autophagosomes maturation into autolysosomes is blocked, only yellow puncta are increased. Thus, we quantified PJ34-induced ratio of red and yellow puncta using chloroquine as positive control of autophagic flux blockage, 24 h treatment with PJ34 confirmed an activation of autophagic flux in comparison with the basal state. In contrast, prolonged treatment with PJ34 (48 h) resulted in a partial inhibition of autophagosomes-lysosomes fusion as suggested by the increased yellow puncta (Figure 2D). Biochemically the autophagy flux could be analyzed as a direct measurement of the formation of the double membrane that makes up the autophagosomes and that finally are going to sequester cytoplasmic content. Concretely the lipidation and translocation of the cytoplasmic protein LC3-I towards the surface of the newborn autophagosome (or LC3-II), was measured and quantified in response to PARPi. Consistent with the Cherry-GFP-LC3 transfection and in combination with lysosomes inhibitors, densitometry quantification of LC3-II revealed an increase in the autophagic flux 24 h after PARPi, followed by a decrease at 48 h after the treatment (Figure 2E).

Interestingly, increased autophagic flux 24 h after PARPi treatment corresponded with increased steady state LC3-II/LC3-I levels, which gives an estimate of the number of autophagosomes in the cell, while the rate of fusion of autophagosomes and lysosomes was unchanged, suggesting that autolysosomes are resolved and the autophagic flux was completed (Appendix A). At 48 h, no increase in autophagic flux was observed (Figure 2E) which corresponded with a decreased rate of fusion and derived in an accumulation of autophagosomes, reflected with the increase in the steady state LC3II/I levels, and suggesting a suppression of autophagosome-lysosome fusion (Appendix A). Similar results were obtained with the GBM cell line U87MG (Appendix A).

### 3.2. PARP Inhibition Induces Lipid Droplets (LDs) Formation and Utilization

It is now broadly accepted that tumor aggressiveness associates with an increased ability to synthesize lipids, providing the necessary resources for survival and rapid proliferation [13,21,22], and fatty acid oxidation is present and active in glioma tissues [14]. Over-expression of lipogenic enzymes has been reported as a common feature in GBM development and tumor progression, [23,24]. In GBM lipid levels and autophagy activation can be interpreted as a stress response orchestrated by the cell under unfavorable conditions. When we analyzed LDs formation as a hallmark of autophagy after PARPi treatment we realized that PARP inhibitors induced LDs synthesis very rapidly and much earlier than other autophagy hallmark, suggesting a specific effect on lipid metabolism (Appendix A).

LDs are intracellular deposits of lipid esters (triglycerides (TGs) and cholesterol) that play an essential role as energy reservoirs. To this end, we stained LN229 and U87MG cells with the lipid marker Sudan Red after PJ34 treatment (Figure 3A and Appendix A), the novel PARP inhibitor BMN-673 or PARP-1 knock-down (Appendix A), and in all cases PARPi induced LDs synthesis. PARPi-induced LDs formation was also observed in SW1783 cell line (anaplastic astrocytoma) as well as in the primary GBM cells MSO4, confirming that LDs accumulation in the absence of PARylation is not a cell line-dependent event (Appendix A).

As shown in Figure 2E, autophagy is activated after blunting PARP activity. Although the release of LD-stored TG and cholesterol has traditionally been attributed to cytosolic lipases, autophagy has been shown to degrade LDs through a pathway termed lipophagy. To test the contribution of lipophagy to LDs turnover in our model, we knocked-down the key autophagy core protein ATG1 which triggered the initial steps in the autophagosomes formation and previous to LC3-II translocation. Specific siRNA ATG1(siATG1) induced the accumulation of LDs (Figure 3B), thus confirming that LDs were accumulated after blunting autophagy in GBM in the basal state and suggesting a role for lipophagy in the basal lipid turnover. We next asked whether LDs produced by PARPi were also used to fuel the cell. Indeed, PARPi treatment further accumulated LDs in siATG1 cells indicating that PARPi-induced autophagy is, at least in part, connected with LDs consumption (Figure 3B). What is the fate of PARPi-induced LDs? Are they consumed to fuel glioma cells? As a first approach to answer this question we performed a double labelling indirect immunofluorescence of the lysosome marker LAMP1 and Bodipy (that stains neutral lipids) in the presence or absence of lysosome inhibitors. Figure 3C confirmed the results obtained after ATG1 knock-down that glioma cells display a basal lipophagy and PARPi induced an accumulation of LDs in autophagosome, but did not increase after inhibition of autophagy, suggesting that LDs induced by PARPi are degraded but not necessarily through lipophagy. Overall, these results suggest that PARP inhibition is contributing not only to LDs formation (Figure 3A and Appendix A) but also to their turnover and utilization (Figure 3B,C).

### 3.3. PARPi-Mediated Suppression of AMPK Modulates LDs Formation

Next, we aimed to unravel the mechanism underlying PARPi-induced LDs formation. Two pathways are involved in the accumulation of free fatty acids (FFA) in the cell, which may then be used for LDs formation: (i) The Phospholipids degradative pathway, that requires cytosolic Phospholipase A2 (cPLA2) to cleave phospholipids and yield FFA, and (ii) De novo lipogenesis, that is initiated with the production of Malonyl-CoA from Acetyl-CoA by the action of the rate-limiting enzyme Acetyl-CoA Carboxylase (ACC).

First, we analyzed whether PARP inhibition modulated the expression of active (phosphorylated) cPLA2. Different times of treatment revealed no activation of cPLA2 indicating that the phospholipids degradative pathway is not involved in PARPi-induced LDs formation lipogenesis (Appendix A).

Next, we tested whether PARPi modulated the de novo lipogenesis through ACC activation (Ser79). In response to low ATP levels, AMPK activation inhibits energy-consuming pathways including FA synthesis through the inhibitory phosphorylation of ACC. Moreover, ROS-induced DNA damage activates PARP and depletes ATP levels, thereby up-regulating AMPK which in turn inactivates ACC [25,26]. Interestingly, increased ROS release has been reported in glioma initiating cells [8], and the efficiency of PARP inhibition in targeting GBM is partly mediated by their ability to decrease ROS levels. In view of this connection, we hypothesized that high basal ROS levels will activate AMPK through ATP decrease and as consequence of ROS-induced DNA damage and PARP over-activation; then, PARP inhibition would prevent AMPK activation (Figure 4A). This hypothesis was confirmed in Figure 4B as PARPi down-regulated AMPK (Thr172) and ACC (Ser79) phosphorylation, which leads to ACC activation. Moreover, the presence of and external source of lipids such as oleic acid (OA) prevented PARPi-derived decreased phosphorylation of ACC and AMPK (Appendix A). ROS scavengers N-acetyl cysteine (NAC) and TROLOX partially abrogated AMPK and ACC phosphorylation (Figure 4B and Appendix A). On the contrary, AICAR (an analog of AMP) induced AMPK and ACC phosphorylation. Indeed, NAC and PJ34 each in-creased intracellular ATP levels (Figure 4C), and promoted LDs formation (Figure 4D and Appendix A). Altogether, these results show that ATP depletion promotes AMPK activation in GBM cells while PARPi prevented ATP drop, driving ACC activation. Furthermore, silencing of ACC completely prevented PARPi-induced LDs formation (Figure 4E).

Together with ACC, another well-known lipogenic enzyme that participates in the de novo FA synthesis is fatty acid synthase (FASN). To better confirm that this pathway is involved in LD formation after PARP inhibition, we analyzed FASN mRNA levels after PJ34 treatment by qPCR (Figure 4F). As expected, FASN expression increased after PJ34 treatment. Altogether, these results confirmed that PARPi modulates de novo lipogenesis.

### 3.4. Lipids from LDs as Metabolic Fuel for Survival of Glioma Cells

To further examine the implication of FA in the resistance of glioma cells to PARP inhibition, we analyzed the effect of PJ34 on cell proliferation, combined or not with oleic acid (OA). The results in Figure 5A show that in the presence of OA, PJ34 significantly increased cell viability; interestingly, in the presence of lysosomal inhibitors, PJ34 and OA failed to increase cell viability (Figure 5A). To further unmask the effect of FA consumption in the resistance to PARP inhibitors, we treated GBM cells with the β-oxidation inhibitor Etomoxir, which potentiated PJ34-induced cell death analyzed as SubG1 fraction (Figure 5B). These results suggest that autophagy and lipids turnover is involved in the acquisition of resistance of GBM cells to PARP inhibition.

To determine whether PARPi-induced LDs served as fuel, we measured oxygen consumption rates (OCR) using Seahorse respirometry. Cells were treated with PARPi combined or not with lysosomal inhibitors or/and Etomoxir. While PJ34 alone had no effect on OCR, the combination with Etomoxir strongly reduced this parameter (Figure 5C,D). Moreover, this combination was the most effective in reducing maximal respiration (Figure 5E). The spare respiratory capacity indicates the capability of the cell to respond to an energetic demand as well as how close cell is to its theoretical maximum respiration and can be used as indicator of cell fitness or viability. Spare respiratory capacity was calculated by subtracting the basal OCR from the maximum OCR after FCCP addition. Interestingly, PJ34 in combination with Etomoxir completely blunted spare respiratory capacity (Figure 5F), corresponding with a strong reduction in cell fitness which is a hallmark of loss of cell viability. In summary, PARPi alone does not have a significant effect in maximal respiration nor spare respiratory capacity. However, its combination with drugs that compromise lipid metabolism such as Etomoxir or lysosomal inhibitors strongly potentiate PARPi effect in these parameters.

## 4. Discussion

We have recently reported that PTEN-deficient glioblastoma cells were particularly sensitive to PARP inhibition (PARPi) [12]. In the current study, we further advance in dissecting PARPi effect in GBM. Importantly, we show that PARPi targets pro-survival pathways in GICs population which are responsible for the initiation and recurrence of the disease. Different pathways are affected thereby compromising the GICs phenotype. The AKT/mTOR pro-survival axis is clearly down-regulated in response to PARP inhibitors, which constitutes a brake for GICs proliferation and self-renewal. In addition, the expression of stemness markers such as OCT4, NANOG, and SOX2 is decreased, further confirming that the GICs niche is compromised after PARP inhibition. In spite of this effect, the global efficiency of PARPi in GBM cells is limited. Thus, we aimed to analyze glioma cells traits that limit the efficacy of PARPi as anti-tumor agents. The strategy orchestrated by GBM implies different responses. We have observed that PARP inhibition exerts a double effect: On the one hand, PARP inactivation blunts the AKT/mTOR pathway, resulting in autophagy activation; on the other, by reprogramming their metabolism GBM cells transform a cytotoxic response (PARPi-induced cell death) into a pro-survival signal leading to the accumulation of fatty acids in the form of lipid droplets.

To elucidate the mechanism connecting PARP inactivation with lipid droplets formation we have focused on the de novo FFA synthesis pathway. Previous results from our group have shown that nutrient starvation increased ROS production, leading to DNA damage and PARP activation [26]. The induction of ATP decrease after PARP activation turned “on” AMPK and its nuclear export to activate autophagy in the cytosol [26]. Interestingly, several evidences suggest that ROS production is constitutively elevated in GBM, without the requirement of any triggering signal like nutrient starvation: first, EGFR is a frequent mutation in glioma and its constitutive activation leads to ROS production [27] and ROS can also directly phosphorylate signaling proteins and increase flux through PI3K/AKT and MAPK/ERK to potentiate oncogenic signaling [28]; second, increased ROS levels in GICs contribute to the oxidative base damage and single-strand DNA breaks found in GICs [8]. Interestingly, increased ROS in GICs is associated with reduced self-renewal, increased cycling and reduced viability through activation of p38 under oxidative stress [29].

We hypothesized that PARP inhibition is contributing to de novo FA synthesis by counteracting the ATP drop generated by DNA damage-induced PARP activation, thereby allowing the activation of ACC. As shown in Figure 3, Figure 4, and Appendix A, both PARP inhibition, NAC and TROLOX (that prevents ROS release), increased ATP levels, leading to inactivation of AMPK, activation of ACC, and the accumulation of LDs. Moreover, knock-down of ACC completely prevented PARPi-induced LDs accumulation. Increased expression of FASN in glioma cells is associated with increased malignancy. Further confirmation of the contribution of PARPi to de novo FA synthesis was obtained by the fact that PARPi triggered FASN expression. Globally, this very early response prepares the cells to acquire subsequent mechanisms to re-set their metabolic needs, including pro-survival autophagy and reliance on FA oxidation to fulfill their energy requirements.

Consistent with these results, recent publications from different groups have demonstrated that glioma cells primarily use FA as a substrate for energy production [14,30,31,32].

Moreover, our results also show that metabolic adaptation not only affects glioma proliferation but confers the cell with pro-survival mechanisms against PARP inhibitors; interestingly, we found that the combination with Etomoxir (a specific and irreversible inhibitor of carnitine palmitoyl transferase I (CPT1), the rate-limiting step in β-oxidation), inhibits respiration and growth of glioma cells, leading to a complete loss of cell fitness. Indeed, Etomoxir has been shown to provide a new therapeutic option for slowing down tumor growth by reducing cellular catabolic activity [14].

Fatty acids are stored in the form of LDs that serve as energy reservoirs to support survival when energy supplies are high. Degradation of lipids stored in LDs can occur to meet the energy needs of the cell. Neutral lipolysis and autophagy are two pathways the cell uses to metabolize fat stored in LDs [33]. The relationship between autophagy and cancer is complex and unclear. Autophagy is reported to show both positive and negative effects on tumor progression [34]. Lipophagy also plays a dual pro- and anti-cancer role, but apart from this, lipophagy-dependent degradation of lipids may provide the rapidly proliferating glioma cells with energy substrates and intermediates for synthesis of biomolecules, thus supporting their survival [35].

As a whole and concluding, our findings suggest that PARP inhibition may be a promising strategy against GBM, as PARPi decreases cell viability [12] down-regulates pro-survival pathways and importantly, compromises the GICs population stability (Figure 6). However, glioma cells convert PARP inhibition-derived cytotoxicity into a pro-survival signal by means to reprogram their metabolism and adapting the autophagy response. These adaptation mechanisms may serve as potential target for glioma therapy and suggest that β-oxidation inhibitors and probably lysosomal inhibitors (Figure 5A) may potentiate the effect of PARP inhibition in glioblastoma, and contribute to increase life expectancy in the patients of this malignant disease.

## 5. Conclusions

As a whole and concluding, our findings suggest that PARP inhibition may be a promising strategy against GBM, as PARPi decreases cell viability [12], down-regulates pro-survival pathways, and importantly, compromises the GICs population stability (Figure 6). However, glioma cells convert PARP inhibition-derived cytotoxicity into a pro-survival signal by means to reprogram their metabolism and adapting the autophagy response. These adaptation mechanisms may serve as potential target for glioma therapy and suggest that β-oxidation inhibitors and probably lysosomal inhibitors (Figure 5A) may potentiate the effect of PARP inhibition in glioblastoma, and contribute to increased life expectancy in the patients of this malignant disease.

## Figures and Tables

**Figure 1 cancers-14-00726-f001:**
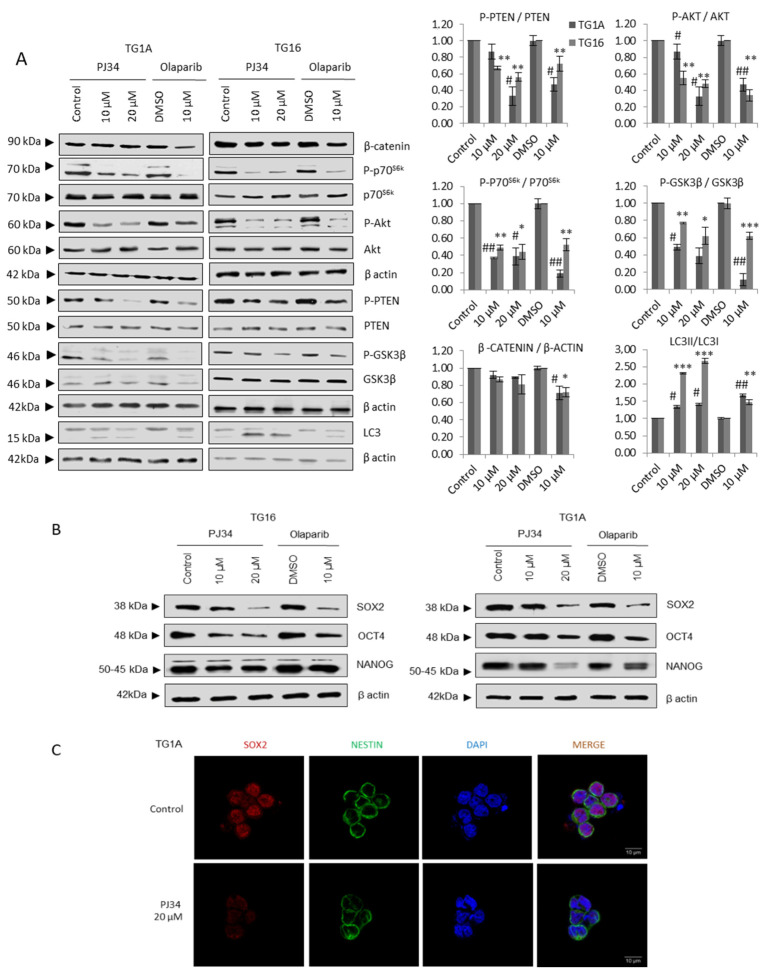
PARP inhibition compromises glioma stem-like phenotype. (**A**) Immunoblots (IB) and graphical representation for the indicated proteins in TG1A and TG16 Glioma Initiating Cells treated with PARP inhibitors PJ34 or Olaparib for one week. (**B**) IB of SOX2, NANOG and OCT4 in TG1A and TG16 cells treated with PJ34 for one week. Full Western Blot images can be found in Appendix A. (**C**) Immunofluorescence (IF) of SOX2 (red) and NESTIN (green) in TG1A cells treated with PJ34 for one week; at least 10 fields of view per condition and replicate were analyzed and quantified. Data are representative of 3 independent experiments. Values are mean ± SEM. * *p* < 0.05, ** *p* < 0.01, *** *p* < 0.001 referred to TG16 and # *p* < 0.05, ## *p* < 0.01 referred to TG1A. Student’s *t* test.

**Figure 2 cancers-14-00726-f002:**
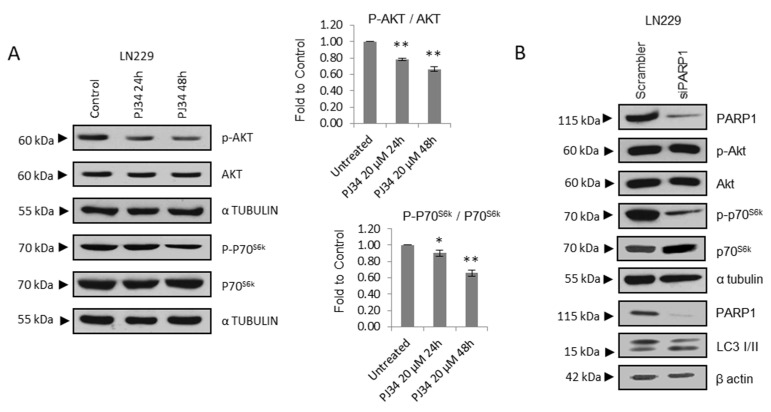
PARP inhibition modulates mTOR pathway and autophagic flux. (**A**) IB and graphical representation for the indicated proteins in LN229 cells treated with PJ34 at the indicated doses and times. (**B**) IB and graphical representation for the indicated proteins in LN229 cells transfected with scramble siRNAs or siRNA against PARP-1. (**C**) Representative electron micrographs depicting autophagosomes (indicated by black arrows) in LN229 and U87MG treated with Olaparib or PJ34 at the indicated doses for 48 h. (**D**) Representative images and graphical representation depicting autophagosomes (yellow) and autolysosomes (red) in LN229 cells transfected with mGFP-LC3-Cherry plasmid and treated with PJ34 or CQ at the indicated doses and times. At least 250 cells were evaluated in 10 independent fields of view per condition. (**E**) IB of LC3 in LN229 cells treated with PJ34 at the indicated doses and times, in the presence or absence of lysosomal inhibitors (Lys Inh) for 4 h. Full Western Blot images can be found in Appendix A. Graphical representation of net LC3-II flux, directly proportional to autophagosomes membrane formation and closure, is shown. Data are representative of at least 3 independent experiments. Values are mean ± SEM. ns.: non statistic, * *p* < 0.05, ** *p* < 0.01. Student’s *t* test.

**Figure 3 cancers-14-00726-f003:**
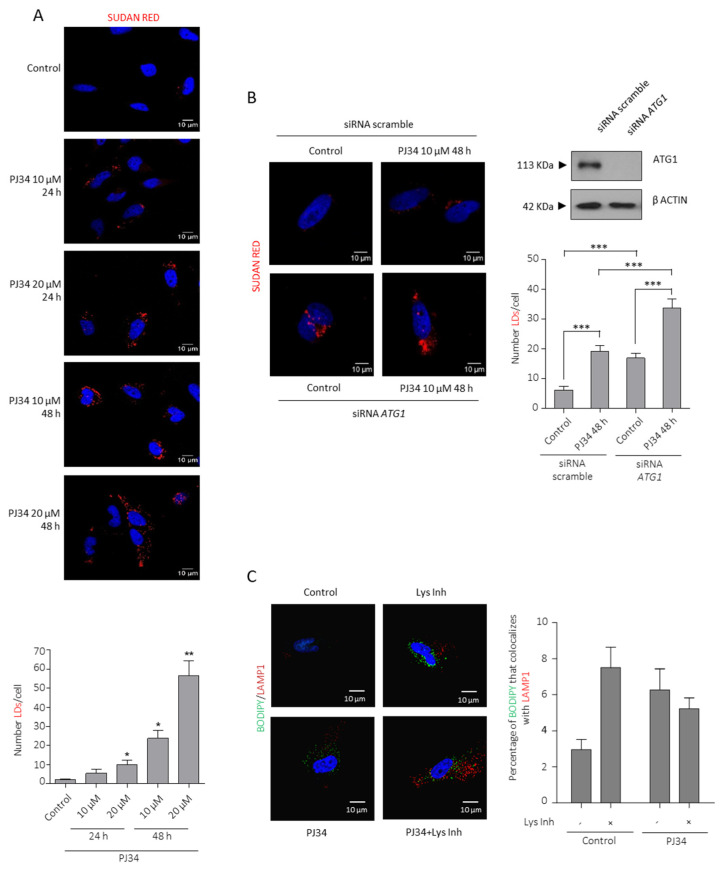
PARP inhibition induces Lipid Droplets formation and utilization. (**A**) Sudan red staining in LN229 cells treated with PJ34 at the indicated doses and times. Lipid droplets quantification using Image J is represented in the right panel. (**B**) Sudan reds staining in LN229 after ATG1 knock-down followed by PJ34 treatment at the indicated dose and time. Lipid droplets quantitation is shown in middle panel. IB for ACC is shown in right panel. Full Western Blot images can be found in Appendix A. At least 150 cells in 5 fields of view per condition, in 3 independent experiments, were quantified in (**A**,**B**). (**C**) IF and graphical representation depicting Bodipy (green) and LAMP1 (red) Colocalization in LN229 cells treated with PJ34 10 µM (48 h) in the presence or absence of Lys inhibitor for 4 h; at least 250 cells in 10 non-related fields of view per condition and replicate were analyzed and quantified by confocal microscopy. All the data are representative of 3 independent experiments. Values are mean ± SEM. * *p* < 0.05, ** *p* < 0.01, *** *p* < 0.001. Student’s *t* test.

**Figure 4 cancers-14-00726-f004:**
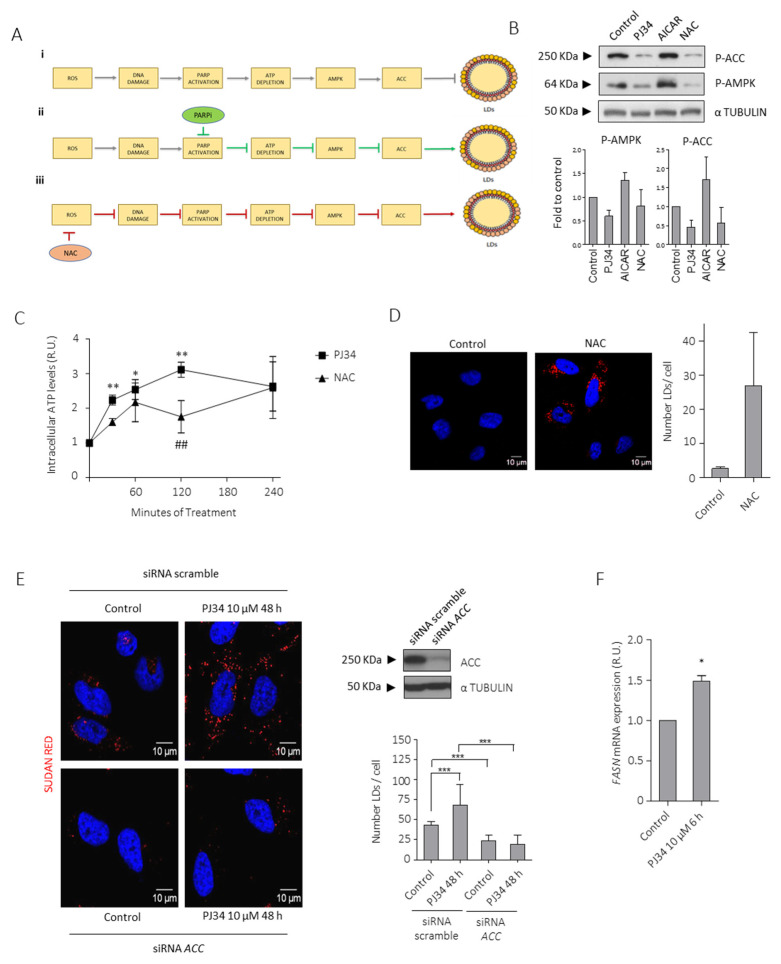
PARPi-induced inhibition of AMPK modulates lipid droplets (LDs) formation. (**A**) Schematic representation of the proposed mechanism that modulates LDs formation in glioblastoma. The effect of PARP inhibition on PARP, ATP, and AMPK activity (ii) (green arrows) and NAC on ROS accumulation, DNA Damage and PARP activation (iii) (red arrows) is also shown. (**B**) IB and graphical representation for phospho-ACC (inactive form) and phospho-AMPK (active form) in LN229 cells treated with PJ34 10 µM, AICAR (AMPK activator) 2 mM or NAC (ROS inhibitor) 3 mM for 4 h. (**C**) Intracellular ATP levels in LN229 cells treated with PJ34 10 µM or NAC 3 mM at the indicated times. (**D**) Sudan red staining in LN229 cells treated with NAC 3 mM for 48 h. Lipid droplets quantitation is shown in right panel. (**E**) Sudan red staining in LN229 cells transfected with scramble siRNA or siRNAs against ACC, in the presence or absence of PJ34 for the indicated dose and time. Lipid droplets quantitation is shown in middle panel; at least 150 cells in 5 fields of view per condition, in 3 independent experiments were quantified in (**D**,**E**). IB for ACC is shown in right panel. Full Western Blot images can be found in Appendix A. (**F**) mRNA levels of FASN in the presence/absence of PJ34 at the indicated dose and time. Data are representative of 3 independent experiments. Values are mean ± SEM. * *p* < 0.05, ** *p* < 0.01, *** *p* < 0.001, ## *p* < 0.01 referred to NAC treatment and ROS accumulation. Student’s *t* test.

**Figure 5 cancers-14-00726-f005:**
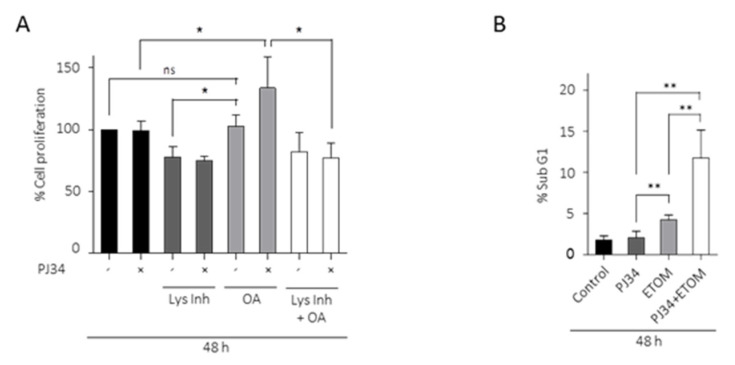
LDs are required as metabolic fuel in glioma cells. (**A**) Proliferation analysis by MTT assay of glioblastoma cells treated with PJ34 10µM (48 h), OA (12 h) or Lys Inh (4 h). Data from 3 independent experiments were expressed as a percentage of the control. (**B**) SubG1 fraction was analyzed by flow cytometry after treatment with PJ34 10 µM and/or Etomoxir 100 µM, in absence of serum, during 48 h. Graph shows the mean of 3 independent experiments, 2 replicates by condition. (**C**,**D**) Oxygen consumption rates (OCR) in LN229 cells using Seahorse respirometry. Baseline measurements were taken for cells in plain medium, which had been exposed to 10% serum 72 h prior to experiment, treated with PJ34 (48 h) and/or Etomoxir (48 h) and/or Lys Inh for 4 h. Cells were then treated with Oligomycin (1.25 µM), FCCP (2 µM) and antimycin A (1.8 µM). (**E**) Maximal respiration, calculated by subtracting the non-mitochondrial respiration from the maximum rate measurement after FCCP injection, is shown. (**F**) Spare respiratory capacity, calculated by subtracting basal respiration from maximal respiration, is shown. Values are mean ± SEM of 5 replicates in one run. Values are mean ± SEM. ns: non statistic, * *p* < 0.05, ** *p* < 0.01, *** *p* < 0.001. Student’s *t* test.

**Figure 6 cancers-14-00726-f006:**
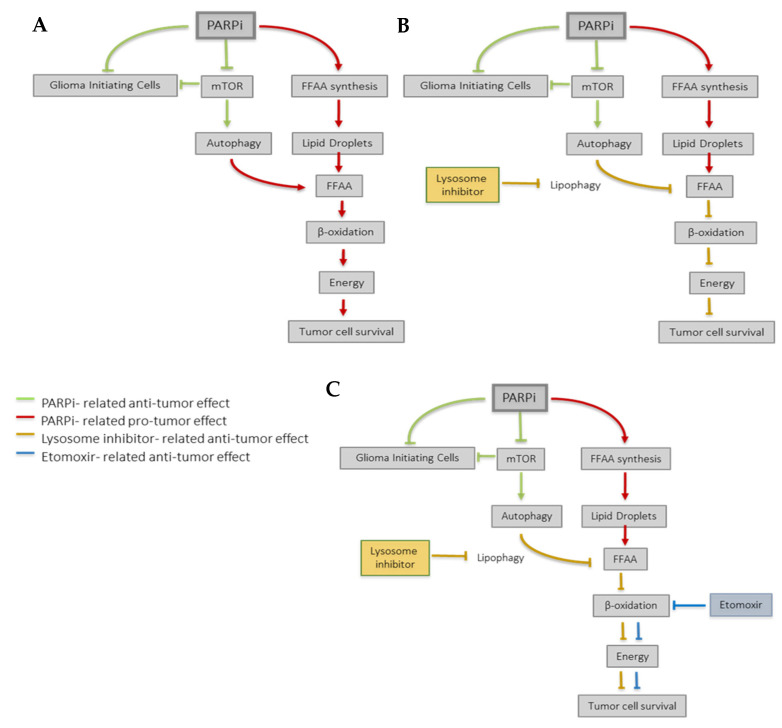
Proposed model for PARPi effect in GBM. (**A**) PARP inhibition exerts different anti-tumoral effects that compromise the Glioma Initiating Cells’ niche and the AKT/mTOR axis (green lines). However, GBM cells also undergo a metabolic reprogramming in response to PARP inhibition (red lines) which contributes to tumor survival. Briefly, PARPi induces the synthesis of fatty acids (FA) that can be stored in lipid droplets. In case of an energetic demand, autophagy promotes the release of FA from lipid droplets, which undergo β-oxidation and contribute to cell survival. (**B**) Lysosomal inhibitors (orange lines) may counteract the pro-survival metabolic reprogramming induced by PARP inhibition and (**C**) this effect may be reinforced by the use of the β-oxidation inhibitor Etomoxir (blue lines), thereby further sensitizing GBM cells to PARPi.

## Data Availability

The data presented in this study are available in this article and Appendix A.

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
