# Peer review of "Glioblastoma Cells Counteract PARP Inhibition through Pro-Survival Induction of Lipid Droplets Synthesis and Utilization"

_cancers, 2022, doi:10.3390/cancers14030726_

Round 1
Reviewer 1 Report
The authors found that suppression of PARP activity exerts an anti-stemness effect and had a dual impact on autophagy, and preventing autophagosomes fusion to lysosomes. In parallel, PARPi triggered the synthesis of lipid droplets (LDs) through ACC-dependent activation of de novo fatty acids (FA) synthesis. They found that blocking FA utilization by inhibition of b -oxidation, increased PARPi-induced glioma cell death while treatment with oleic acid (OA) prevented the anti-glioma effect of PARPi.
Major revision
Only one cell line (LN229) and only one PARPi (PJ34) were used in Figure 3-5.
Other cell line and other PARPi should be used to confirm the results.
Minor revision
Figure 2
Figure 2a-c was not shown.
Figure 5A
There was no figure showing the cell proliferation data of cells treated with ETOM or ETOM + PJ34. The data should be added to Figure 5A.
Reviewer 2 Report
This paper explores how Giblastomas respond to PARP inhibition with molecular inhibitors (PARPi )in different ways. The authors present evidence of how PARPi displays an anti tumorigenic effect by exerting anti– cancer stemness effect, there is also a disruption of autophagy pathways. The manuscript interestingly provides evidence how Giblastomas can overcome PARPi by the induction of Lipid droplets by reprogram their metabolic pathways. The study concludes to provide insight into potential combination therapies that target lipid metabolism. The study on the whole would be of interest to the readership of this journal and shows some novel data that would be interest to the cancer research community. I have a few concerns that would need addressing regarding RNAi experiments, and how data is quantified regarding the autophagy and Lipid droplet experiments. The study is also somewhat confusing to follow (see below figure 2 ) with some data displayed in supplementary information that need to be in main figure this would require improvement.
Major points
1
I can see from the text in line 263 that inhibitors were added for 1 week and re-added during this week again after 3 days. This seems like a long time after to look at cellular signalling pathways associated with phosphorylation of proteins, perhaps the authors may have missed something that is occurring at shorter additions of these inhibitors?
2
Figure 2A, B and C panels missing ? found data in Figure S2 this make it very confusing to read please correct this.
Figure 2/Sup figure 2, panel A. The authors claim that RNAi mediated inhibition give same profile regarding reduction of p-AKT and LC3 I/II levels as PARP inhibitors, however the inhibition of p-AKT seems only slightly reduced when compared to figure 1 A and does not seem very convincing.
3
The graphical quantitation of the RNAi in Sup figure 2 show small reduction in p-AKT however the blots are far from convincing of this effect please could the authors replace these blots with more representative examples.
4
Why is the TEM data not made into a figure panel? and should this data not be quantified correctly to show the increase number of autophagosomes ?
5
Figure 2 D Please could indicate where there are red vesicles in the images I can only see yellow only ? which is inconsistent with the quantitative graph showing approximately 10 fold increases in red vesicles. Could the authors show for clarity the individual channels for red and green in grey only.
6
Figure 3, Figure S4 figure 4 E please could the authors show for clarity the individual channels for red of the Sudan red staining.
Figure 3 Looking from the immunofluorescent images it is not clear that in control cells there is any staining at all, yet the quantitation measures 5-10 number of lipid droplets /cell yet in figure 4 E we do see in control image cells with sudan red staining.
IFigure 3 and 4 E. It would be good to show graph showing the % of cells in control vs inhibitor/siRNA that have any sudan red staining as it is somewhat misleading to display this as a LD/cell. Did this data come from one cell or many cells ? The authors should show larger field of view of cells to demonstrate this and indicate how many cells they are measuring number of LD/cell if they want to quantify the data in this way.
Minor points
1
figure 1 A Quantitation graphs of the blots require more detail, how this was performed, were bands first normalised to b-Actin bands and then normalised to control ? Molecular weight markers are missing on figure S1A, figure 3 B, figure 4 E blot figure S5 A
2
Arrows are required to indicate molecular weight markers on all immunoblots.
3
Figure 1 A The graph are missing any significance values. And missing labels for PI34 and Olaparib inhibition.
4
Figure1 B Any comments on why TG1A cell line responds differently in nanog blot?
5
Please remove from line 471-473 I’m guessing this was added in error
Reviewer 3 Report
This manuscript by Majuelos-Melguizo et al reports an interesting finding, namely, how PARPi-treated GBM cells can mount a salvage pathway of forming lipid droplets. The findings sound novel and are presented in a logical way. Overall, I have no objection in publishing this manuscript in Cancers, for it provides us with an adequate level of novelty and thoroughness. However, I would like to point out a couple of things to improve the current manuscript as follows:
- In Figure2, I cannot see panels A and B, yet the legend says them. Why does the figure starts with D, but the legend starts with A. The authors should clarify this.
- In Figure 2E or what they intended to say, would you tell us how the LC3 influx is defined. I think I know what you meant, but for the sake of readers’ comprehension, it should be spelled out both in the figure legend and in the main text.
- Regarding immunofluorescence quantification, would you indicate in the figure legend how many cells were taken into consideration in making the graphs and statistical testing?
- Abstract is a bit too general and too short to summarize the main text. I recommend elaborating the current Abstract with more detailed information.
- Lack of a figure in this manuscript would draw less attention from readers. While Tables are excellent, having (a) figure(s) would enhance readers’ comprehension.
- Diagrams in Figure 4A and Figure 6 look extremely hard to follow. It may be due to the multiple lines with different colors that connect two points, one representing a negative relation while another a positive regulation, etc. Would you come up with a simpler drawing for both Figure 4A and Figure 6?
Round 2
Reviewer 1 Report
The paper is acceptable for publication.
Reviewer 2 Report
The authors have done well in addressing all my concerns and have produced a much improved manuscript